# Study of Consumer Liking of Six Chinese Vinegar Products and the Correlation between These Likings and the Volatile Profile

**DOI:** 10.3390/foods11152224

**Published:** 2022-07-26

**Authors:** Shan Liang, Ying Liu, Shao Yuan, Yixuan Liu, Baoqing Zhu, Min Zhang

**Affiliations:** 1Beijing Engineering and Technology Research Center of Food Additives, Beijing Technology and Business University, Beijing 100048, China; liangshan@btbu.edu.cn (S.L.); lyly0817@126.com (Y.L.); 2College of Biological Sciences and Technology, Beijing Forestry University, Beijing 100083, China; shaoyuan1998@163.com (S.Y.); lyxlyxlyx2014@126.com (Y.L.)

**Keywords:** Chinese vinegar, gas chromatography–mass spectrometry (GC–MS), volatile compounds, sensory evaluation, open-ended questions

## Abstract

As the aroma of Chinese vinegar is a key quality trait that influences consumer liking, a combination of sensory data and instrumental measurements were performed to help understand the aroma differences of six types of Chinese vinegar. A total of 52 volatile compounds, mostly ethyl acetate, acetic acid, and phenethyl alcohol, were detected in six types of Chinese vinegar using solid-phase microextraction coupled with gas chromatography–mass spectrometry (SPME-GC–MS). Combined with open-ended questions, the correlation between consumer liking and the volatile profile of the vinegar was further investigated. More consumers preferred the potato vinegar (B6) described as “having a sweet aroma and fruity vinegar aroma”. The Heng-shun Jinyou balsamic vinegar (B5) was not favored by consumers with its exhibition of “too pungent vinegar aroma”. Based on their preference patterns, consumers were grouped into three clusters by k-means clustering and principal component analysis (PCA). Using partial least squares regression (PLSR), the most important volatile compounds that drove consumer liking in the three clusters were obtained, among which 14 compounds such as 1-methylpyrrole-2-carboxaldehyde, ethyl acetate, and acetylfuran had the greatest impact on consumer liking, which could guide manufacturers to improve product quality and customer satisfaction.

## 1. Introduction

Vinegar is a popular seasoning and cooking ingredient that contains acetic acid and other flavor components [1]. The acetic acid in vinegar is mainly produced from ethanol via acetic acid fermentation [2]. Vinegar has attracted increasing attention since various studies have suggested that its consumption can improve human health [3]. For instance, vinegar reportedly contains many nutritional ingredients, including amino acids, minerals, organic acids, and phenolic compounds, displaying anti-microbial and anti-oxidant properties that can prevent hypertension, cardiovascular disease, and cancer [3,4,5,6,7,8].

Since Chinese vinegar is used extensively, the country produces 26 million hectoliters of vinegar annually, and its quality is essentially determined by its appearance, aroma, and nutritional components [9]. Volatile compounds play a vital role in determining the overall aroma of Chinese vinegar and are mainly formed from the source material (grain and cereal) via the fermentation process [9,10,11]. It has been reported that different acetic acid bacterial strains possess different preferences on metabolizing nutrient components during fermentation, which could result in different aromatic features for Chinese vinegar [12]. After the fermentation process, Chinese vinegar is aged to enhance its sensory attributes and nutritional quality while improving the aromatic complexity [13,14,15].

Solid-phase microextraction (SPME) represents a well-established sampling method, which can be used to extract many volatile compounds from a large variety of foods. It has been frequently combined with gas chromatography–mass spectrometry (GC-MS), which is widely used to identify volatile compounds [16,17]. Chung et al. (2017) [18] reported that SPME-GC–MS helped to distinguish the aroma profiles of rice vinegars of different producer origin, reflecting the important role of SPME-GC–MS in the extraction and identification of volatile compounds.

Regarding sensory analysis, a descriptive assessment by a trained panel represents the typical method used in the food industry to develop and control the sensory quality of products [19,20,21]. However, creating and maintaining well-trained, calibrated sensory panels can be economically challenging and time-consuming, particularly when dealing with a complex product, such as wine [22]. Moreover, due to extensive training, highly trained assessors can perceive wine aroma differently from consumers, who display a unified and holistic impression of the product. Some studies have indicated that the perception of a trained panel does not reflect the sensory impression of consumers [19,21]. Considering the high competitiveness of the current market, companies must base their decisions on consumer preferences to increase the success of their products [23]. Open-ended questions are a fast sensory descriptive analysis method. Recent studies have employed it to evaluate 3D printed cookies and coffee [24,25]. Applying it to sensory food evaluation not only complements the quantitative results provided by the sensory panels and helps to explore the similarities and differences between products but also provides considerable information for product developers and designers.

To better understand vinegar aroma perception, finding a correlation between the sensory data and instrumental measurements is necessary [26]. The combination of sensory data and instrumental measurements helped facilitate marketing and quality control. Yu et al. (2021) [27] revealed the aroma characteristics of traditional Huangjiu produced around the winter solstice via sensory evaluation, GC–MS, and gas chromatography–ion mobility spectrometry (GC–IMS). The results suggested that the traditional Huangjiu produced around the winter solstice contained more aroma volatile compounds and had better aroma quality than those produced during other periods. It proved that the combination of sensory data and instrumental measurements could guide product optimization effectively. The present study selected six different types of commercially available Chinese vinegar for volatile compounds extraction and analysis using SPME-GC–MS while exploring the association between consumer perception and volatile composition. This study aims to characterize the aromatic features of these vinegar samples to help understand the relationship between volatile compounds and sensory attributes, and guide manufacturers to improve the quality and consumer liking of vinegar.

## 2. Materials and Methods

### 2.1. Chemicals

The external standards with a purity of at least 95% included ethyl acetate, diethyl succinate, isoamyl acetate, benzaldehyde, isovaleric acid, caproic acid, octanoic acid, propionic acid, phenylethyl alcohol, and were purchased from Sigma-Aldrich (St. Louis, MO, USA). Furthermore, 2-methyl-3-heptanone with a purity of 99% represented the internal standard and was also obtained from Sigma-Aldrich (St. Louis, MO, USA).

### 2.2. Chinese Vinegar Samples

In this study, six representatives of different types of vinegar produced in various regions of China were selected, including ten-year aged Qian-he cellar vinegar (B1), Ning-hua-mansion old vinegar (B2), East-lake health vinegar (B3), Qian-he glutinous rice vinegar (B4), Heng-shun Jinyou balsamic vinegar (B5), and potato vinegar (B6). Among them, potato vinegar (B6) was selected because potato (one of the principal raw materials) is widely cultivated around the world, is rich in nutrients, and has enormous development potential. All the above samples were purchased from a local supermarket (Beijing, China). Detailed information regarding these Chinese vinegar samples is listed in Table 1.

### 2.3. Volatile Compounds Extraction

The volatile compounds were extracted from the Chinese vinegar samples using SPME according to a published method with minor modifications [28]. Briefly, each Chinese vinegar sample (5 mL) was mixed with 1 µL of 0.816 µg/µL 2-methyl-3-heptanone and 1 g sodium chloride in a 15-mL vial containing a magnetic stirrer. The vial was immediately capped with a PTFE-silicone septum and equilibrated in a 55 °C water bath under agitation for 20 min. Next, a DVB/CAR/PDMS fiber was inserted into the headspace of the vial to adsorb the volatile compounds for 40 min at the same temperature with the same agitation (55 °C water bath under agitation). After SPME, the fiber was removed from the headspace of the vial and immediately inserted into the injection port of the gas chromatograph; it was then left for 5 min at 250 °C to desorb the volatile compounds into the GC column. All the samples were analyzed in triplicate.

### 2.4. GC–MS Analysis

An Agilent 7890A GC coupled with an Agilent 7000B mass spectrometer (Agilent Technologies, Santa Clara, CA, USA) was used to analyze the volatile compounds in the Chinese vinegar samples according to a previously described method [28]. An Agilent DB-WAX capillary column (30 m × 0.32 mm, 0.25 µm film thickness, Agilent Technologies, Santa Clara, CA, USA) was employed to separate the volatile compounds using a carrier gas (helium) at a flow rate of 1 mL/min. The temperature of the oven was programmed as follows: The temperature was increased from 40 °C to 250 °C at 5 °C/min, and maintained at 250 °C for 3 min. A 5:1 split mode was used under an electron impact mode of 70 eV, with a mass spectrometer interface temperature of 280 °C, and an ion source temperature of 230 °C. A selective ion mode was used for the mass scan, ranging from *m/z* 20 to *m/z* 450. A C6-C24 n-alkane series (Supelco, Bellefonte, PA, USA) was used in the same chromatographic conditions to calculate the retention indices. Volatile compounds with available reference standards were identified by comparing their mass spectra and retention indices with the standard, while volatile compounds without available standards were tentatively identified by comparing their mass spectra and retention indices with the Standard NIST11 library and reference literature [29]. The stock solution (5 mL) was mixed with 5 mL distilled water and then consecutively diluted to six concentration level standards, which were extracted and analyzed employing the same procedure used for the Chinese vinegar samples. A quantitative analysis was carried out through the standard curve. In addition, volatile compounds without a standard curve were quantified using standards sharing similar structures or carbon atom numbers.

### 2.5. Odor Activity Value (OAV)

The OAVs of volatile compounds reflected their importance in contributing aroma notes to the overall aroma of the sample, and were calculated by comparing their concentrations in the sample with their perception threshold [30,31]. Odor thresholds were taken from the literature [32]. A volatile compound OAV higher than 1 indicated that its aroma features significantly contributed to the overall aroma of the sample.

### 2.6. Sensory Evaluation

This survey was conducted in January 2018 and featured 86 healthy participants (67% women and 33% men, aged 18 to 40) from the Beijing Forestry University. The inclusion criteria were regular vinegar consumption, as well as sufficient interest and time to participate in the study. The respondents were asked to complete an online questionnaire before evaluating the samples, which consisted of 32 questions divided into four sections: (1) Basic Information of Consumers. This section included eight questions regarding name, gender, age, permanent residence in the last ten years, occupation, vinegar consumption frequency, experimental participation time, and contact information. (2) Preference for Vinegar. This section investigated the attitude of the respondents towards vinegar via ten statement questions, which were scored on a 7-point scale. The 1–7 scale represented the responses, “strongly disagree”, “disagree”, “disagree slightly”, “indifferent”, “agree slightly”, “agree”, and “strongly agree”, respectively. The respondents were required to rate each statement question according to their actual situations. (3) General Health Interest. This section also investigated the attitude of the respondents towards vinegar using ten statement questions scored on a 7-point scale. The 1–7 scale represented the responses, “strongly disagree”, “disagree”, “disagree slightly”, “indifferent”, “agree slightly”, “agree”, and “strongly agree”, respectively. The respondents were required to rate each statement question according to their actual situations. (4) Consumer Purchasing Behavior and Preferences. This section consisted of four questions regarding understanding the vinegar aroma, the factors valued by the participants when choosing vinegar, the difference in vinegar quality, and the type of vinegar generally purchased.

This study evaluated six vinegar samples during a single experiment lasting for approximately 15–30 min. Here, 5 mL of each sample was placed in separate 30 mL brown PET plastic vials at room temperature and labeled with random three-digit numbers. The samples were placed on separate tables under artificial white light. The participants were required to smell each sample, with a 3 min resting period between samples to remove the residual odor of the previous sample. After entering the evaluation room, the respondents smelled six different vinegar samples successively to determine the difference between the vinegar aromas. Then, the respondents scored each sample within 30 s according to their personal liking and provided comments for 2 min. The vinegar samples were subjected to a sequential blind test, while the smelling order was rotated for each respondent to avoid the bias caused by the smelling sequence. This test was repeated six times. A 7-point scale was used, with points 1–7 representing “particularly dislike”, “dislike”, “dislike slightly”, “just so, so”, “like slightly, “like”, and “particularly like”, respectively. After the evaluation, the respondents were asked to reply to an additional two questions. (1) “Why do you like this sample?” → “Is there any other reason?” (2) “Why don’t you like this sample?” → “Is there any other reason?” The tests were conducted in controlled conditions in accordance with the ISO8589:2007 standard. All respondents have consented to participation in the study. In the consumer stage, there were 86 participants, while only 76 participants completed the sensory evaluation. The data in this study were the data of the 76 participants

### 2.7. Statistical Analysis

The data were expressed as the mean ± standard deviation of triplicate tests. An analysis of variance (ANOVA) was performed to compare the significant differences between the means using Duncan’s range test and SPSS version 23.0 (Chicago, IL, USA) with a significant level of 0.05. In addition, the Kruskal-Wallis test was used to analyze the consumer liking score. Principal component analysis (PCA) was used to evaluate the similarities and differences between the Chinese vinegar samples regarding their volatile compositions and aromatic properties. All statistical analyses of the sensory data were conducted in the R language and employed packages, such as ggplot2, reshape2, FactoMineR, pheatmap, and PLSR. PCA and k-means clustering were used to draw a consumer preference map using the relevant product data. Partial least squares regression (PLSR) was used to investigate the relationship between the volatile compound concentrations and product liking of each consumer cluster. The PLSR data were scaled and centered according to the volatile compound structures.

## 3. Results

### 3.1. Volatile Compounds Detection Using GC–MS

A total of 52 volatile compounds were detected in vinegar thanks to GC–MS analysis, including eleven esters, seven aldehydes, seven acids, four phenols, three alcohols, three ketones, eight furans, three pyrazines, and six others (Table 2). The relevant information of the standard curve for vinegar compounds is provided in Table 3. The GC–MS total ion chromatograms of six kinds of vinegar are in Appendix A.

#### 3.1.1. Esters

The solid-state fermentation of traditional Chinese vinegar favors ester accumulation, substantially improving the aromatic complexity [33]. This study revealed eleven esters in the Chinese vinegar samples, including ethyl acetate, ethyl propionate, n-propyl acetate, isobutyl acetate, isoamyl acetate, 1,2-propanediol,2-acetate, trimethylene acetate, ethyl benzoate, diethyl succinate, ethyl phenylacetate, and β-phenethyl acetate (Table 4). B1 exhibited high 1,2-propanediol,2-acetate (344,314.42 µg/L) and B2, B3, B4, B5, and B6 all displayed high ethyl acetate concentrations of 625,514.35 µg/L, 783,331.79 µg/L, 571,951.58 µg/L, 304,167.86 µg/L, and 467,917.75 µg/L, respectively. Meanwhile, ethyl phenylacetate was the lowest (trace amounts) in all samples.

Esters represent essential volatile compounds providing vinegar with floral or fruity aromas [34]. Ethyl propionate, denoting sweet, fruity, grape, ether, rum, and pineapple notes, contributed most to the aroma of B1 (OAV = 118.177) (Table 5) B2 (OAV = 170.431), and B3 (OAV = 173.433). Ethyl acetate (OAV = 114.390), ethyl benzoate (OAV = 73.764), and trimethylene acetate (OAV = 263.524) were more representative of the aroma of B4, B5, and B6, respectively. The diethyl succinate (OAV = 5.698) concentration only reached an odor threshold in B5, while isobutyl acetate and ethyl benzoate failed to reach an odor threshold in B3.

#### 3.1.2. Acids

Studies had shown that vinegar contains an abundance of acid compounds, primarily acetic acid, which was consistent with the findings of this paper. Acetic acid is produced via the alcoholic fermentation of wine yeast and by acetobacter acting on alcohol. Other acids may be the products of amino acid degradation via oxidation, or the reduction or the oxidation and degradation of saturated fatty acids [35]. Seven acids were present in all the samples, including acetic acid, propionic acid, butyric acid, isovaleric acid, 2-methylbutyric acid, caproic acid, and octanoic acid. Acetic acid was the most abundant in all the samples at levels of 8,020,749.17 µg/L, 7,057,979.52 µg/L, 11,429,688.79 µg/L, 6,119,026.91 µg/L, 11,729,001.22 µg/L, and 6,077,922.73 µg/L in B1, B2, B3, B4, B5, and B6, respectively. Butyric acid was the lowest in B1, B2, B3, B4, and B6 at 73.83 µg/L, 155.64 µg/L, 230.41 µg/L, 46.56 µg/L, and 40.22 µg/L, while isovaleric acid was the lowest in B5 at 793.42 µg/L. Various other acids were distributed between these concentrations.

Acid compounds are vital for providing vinegar with is bold aromas and include strong, acidic, pungent, spicy, cheesy, and chemical notes [36]. These compounds significantly contribute to the overall aroma of the vinegar and lay the foundation for its sour taste [37,38]. Acetic acid is responsible for strong acidic notes, contributing significantly to the aroma of B1 (OAV = 3645.795), B2 (OAV = 3208.173), B3 (OAV = 5195.313), B4 (OAV = 2781.376), B5 (OAV = 5331.364), and B6 (OAV = 2762.692). The 2-Methylbutyric acid is responsible for pungent, spicy, cheesy notes and was second to acetic acid in aroma strength in B1 (OAV = 191.887), B2 (OAV = 119.274), B3 (OAV = 97.302), B4 (OAV = 136.356), B5 (OAV = 233.104), and B6 (OAV = 72.358), followed by caproic acid (OAV = 9.514). The propionic acid concentrations only reached odor thresholds in B2 (OAV = 4.165), B3 (OAV = 1.221), and B6 (OAV = 25.182), while the butyric acid (OAV = 3.679) and isovaleric acid (OAV = 1.133) concentrations only reached odor thresholds in B5. 

#### 3.1.3. Aldehydes

Aldehyde formation may be enhanced by oxidation after a long aging period [39]. Seven aldehydes were present in all the samples, including 3-methylbutyraldehyde, nonanal, benzaldehyde, phenylethanal, 1H-pyrrole-2-carbaldehyde, 5-methyl-2-phenyl-2-hexenal, and 1-methylpyrrole-2-carboxaldehyde. Of these, benzaldehyde may be derived from the oxidation of benzyl alcohol or the action of microorganisms on phenylalanine, phenol, phenylacetic acid, and hydroxybenzoic acid [40]. Phenylethanal is formed via the Strecke degradation of phenylalanine during the acetic acid fermentation stage [41]. Here, 3-methylbutyraldehyde was most abundant in B1, B2, B3, B4, and B5 at levels of 39,262.87 µg/L, 22,572.34 µg/L, 19,246.37 µg/L, 35,396.43 µg/L, and 44,409.15 µg/L, while phenylethanal was the highest in B6 at 12,303.69 µg/L. The 1-methylpyrrole-2-carboxaldehyde levels were lowest in all the samples at 36.40 µg/L, 135.69 µg/L, 489.55 µg/L, 27.11 µg/L, 92.43 µg/L, and 14.36 µg/L in B1, B2, B3, B4, B5, and B6, respectively.

The aldehyde compounds displayed significant diversity and were highly abundant in the vinegar, providing fruity, floral, fatty, waxy, and fragrant aromas [9]. These components substantially affected the overall aroma characteristics of the different types of vinegar. Phenylethanal contributed sweet, roasted, green, nutty, and floral notes, significantly affecting the aroma profiles of B1 (OAV = 4154.525), B2 (OAV = 3420.799), B3 (OAV = 4034.055), B4 (OAV = 5790.261), B5 (OAV = 3526.653), and B6 (OAV = 3075.922). The benzaldehyde concentration contributed fruity, nutty, woody, and floral notes while reaching odor thresholds in B2 (OAV = 1.905) and B4 (OAV = 2.080).

#### 3.1.4. Volatile Phenols

Phenolic compounds are mainly produced by thermal degradation via the depolymerization or oxidation of lignin [9]. Four phenols were present in all the samples, including guaiacol, 2-ethyl-3-hydroxy-4H-pyran-4-one, 4-ethyl-2-methoxyphenol, and 4-ethylphenol. Here, 4-ethyl-2-methoxyphenol was most abundant in B1, B4, and B6 at levels of 1836.67 µg/L, 4746.59 µg/L, and 2794.37 µg/L, respectively, while guaiacol was highest in B2, B3, and B5 at respective levels of 42,367.62 µg/L, 5063.67 µg/L, and 809.65 µg/L. The 4-ethylphenol level was lowest in all the samples at 23.19 µg/L, 2829.93 µg/L, 180.21 µg/L, 14.55 µg/L, 18.91 µg/L, and 30.89 µg/L in B1, B2, B3, B4, B5, and B6, respectively.

Phenol compounds provide medicinal, meaty, woody, fruity, and floral notes [9]. The guaiacol imparted smoky, spicy, fragrant, meaty, woody, and floral notes, reaching odor thresholds in B1 (OAV = 65.961), B2 (OAV = 2017.506), B3 (OAV = 241.127), B4 (OAV = 28.278), B5 (OAV = 38.555), and B6 (OAV = 60.044). Neither 4-ethyl-2-methoxyphenol nor 4-ethylphenol reached an odor threshold.

#### 3.1.5. Alcohols

Alcohols are mainly derived from alcohol fermentation [42]. Three alcohols were present in all the samples, including 3-methyl-1-butanol, 2,3-butanediol, and phenethyl alcohol. The phenethyl alcohol concentrations were most abundant in all the samples at levels of 460,480.92 µg/L, 145,519.63 µg/L, 77,189.78 µg/L, 565,184.73 µg/L, 695,627.22 µg/L, and 173,905.87 µg/L in B1, B2, B3, B4, B5, and B6, respectively. Furthermore, the 3-methyl-1-butanol concentrations were lowest in B1, B2, B4, B5, and B6, displaying levels of 8637.39 µg/L, 440.01 µg/L, 13,010.93 µg/L, 4574.55 µg/L, and 4819.93 µg/L, respectively, while only trace amounts were evident in B3.

Alcohol compounds provide fruity, floral, fatty, fragrant, and floral notes. The phenethyl alcohol presented a soft, pleasant, and long-lasting scent [43], reaching odor thresholds in B1 (OAV = 613.975), B2 (OAV = 194.026), B3 (OAV = 102.920), B4 (OAV = 753.580), B5 (OAV = 927.503), and B6 (OAV = 231.874). Only 2,3-butanediol (OAV = 1.067) reached an odor threshold in B5, while 3-methyl-1-butanol did not reach an odor threshold.

#### 3.1.6. Ketones

Ketone volatile compounds are formed via sugar degradation by the Maillard reaction [44]. Three ketones were present in all the samples, including 3-hydroxy-2-butanone, acetophenone, and 2-pyrrolidinone. Of these, 3-hydroxy-2-butanone, also known as acetoin, is responsible for a milky aroma. 2,3-butanediol and other by-products will be produced when 3-hydroxy-2-butanone is produced by the glycolytic pathway. Many microorganisms including Bacillus, E. coli and Klebsiella, can be synthesized as 3-hydroxy-2-butanone [45]. The accumulation of 3-hydroxyl-2-butanone plays an important role in pyrazine synthesis. The acetophenone concentrations were most abundant in B1, B3, and B5 at respective levels of 1412.14 µg/L, 1057.34 µg/L, and 1220.89 µg/L, while the 3-hydroxy-2-butanone concentrations were highest in B2, B4, and B6 at 6910.25 µg/L, 2468.94 µg/L, 9315.87 µg/L, respectively. The 3-hydroxy-2-butanone levels were lowest in B3 at 136.94 µg/L, while only showing trace amounts in B1. Furthermore, the 2-pyrrolidinone concentrations were lowest in B2, B4, and B5 at 769.83 µg/L, 328.93 µg/L, and 362.42 µg/L, respectively, while the lowest concentration of acetophenone was found in B6 at 129.10 µg/L.

Ketones provide medicinal, balsam, and floral notes [46]. The acetophenone concentrations, presenting strong medicinal, almond aromas, reached odor thresholds in B1 (OAV = 21.738), B2 (OAV = 86.984), B3 (OAV = 16.267), B4 (OAV = 19.342), B5 (OAV = 18.783), and B6 (OAV = 1.986). The 3-hydroxy-2-butanone concentrations, presenting creamy, fatty aromas, reached odor thresholds in B2 (OAV = 49.359), B4 (OAV = 17.635), B5 (OAV = 2.641), and B6 (OAV = 66.542), but not in B1 and B3. Furthermore, the 2-pyrrolidinone concentrations did not achieve odor thresholds in any of the vinegar samples.

#### 3.1.7. Furans

The furans in vinegar are mainly produced by sugar degradation via heating [9]. Eight furans were present in the samples, including furfural, acetylfuran, furfuryl acetate, 1-pentanone, 1-(2-furanyl)-, 3-furanmethanol, 1-(5-methyl-2-furyl)ethan-1-one, 4-(2-furyl)-3-buten-2-one, and 5-acetyldihydrofuran-2(3H)-one. The furfuryl acetate concentrations were most abundant in B1, B4, and B5 at 19,300.71 µg/L, 18,218.52 µg/L, and 12,829.97 µg/L, respectively, while acetylfuran was highest in B2, B3, and B6 at respective levels of 24,330.76 µg/L, 17,231.94 µg/L, and 3161.14 µg/L. Furthermore, B1, B3, B4, and B5 exhibited the lowest 1-pentanone, 1-(2-furanyl) levels at 93.85 µg/L, 122.23 µg/L, 46.35 µg/L, and 2.39 µg/L, respectively, while the lowest 3-furanmethanol concentration was evident in B2 at 787.66 µg/L. B6 displayed trace amounts of furfural, which could be converted into D-glucose via a series of changes.

Furans provide roasted, woody, fruity, floral, fatty, and floral notes [47]. The 5-acetyldihydrofuran-2(3H)-one concentration, presenting a sweet, lemony scent, reached odor thresholds in B1 (OAV = 7.372), B2 (OAV = 19.470), B3 (OAV = 4.672), B4 (OAV = 8.375), B5 (OAV = 9.111), and B6 (OAV = 6.361). The 1-pentanone, 1-(2-furanyl)- reached odor thresholds in B1 (OAV = 15.642), B2 (OAV = 245.359), B3 (OAV = 20.371), and B4 (OAV = 7.726), but not in B5 and B6. The acetylfuran concentrations, presenting baked, smoky aromas, reached odor thresholds in B2 (OAV = 2.433) and B3 (OAV = 1.723). None of the other furan compounds reached aroma thresholds.

#### 3.1.8. Pyrazines

Pyrazines are produced by microbial fermentation or via the Maillard reaction and amino ketone condensation produced by Strecker degradation [48]. Three pyrazines were present in the samples, including 2-methylpyrazine, 2,3-dimethylpyrazine, and 2,3,5-trimethylpyrazine. The 2,3,5-trimethylpyrazine concentrations were highest in all the vinegar samples at 29,125.40 µg/L, 114,532.53 µg/L, 94,445.64 µg/L, 53,390.76 µg/L, 6870.70 µg/L, and 911.37 µg/L in B1, B2, B3, B4, B5, and B6, respectively. Additionally, all the samples exhibited the lowest 2,3-dimethylpyrazine concentrations at respective levels of 297.71 µg/L, 3190.07 µg/L, 1498.79 µg/L, 627.20 µg/L, 91.46 µg/L, and 45.30 µg/L in B1, B2, B3, B4, B5, and B6.

Pyrazines provide roasted, fragrant, mildewy, ester, and floral notes [32]. The 2,3,5-trimethylpyrazine concentrations, presenting baked, nutty, mildewy, and earthy aromas, reached odor thresholds in B1 (OAV = 16.181), B2 (OAV = 63.629), B3 (OAV = 52.470), B4 (OAV = 29.662), and B5 (OAV = 3.817), but not in B6. Neither 2-methylpyrazine nor 2,3-dimethylpyrazine reached odor thresholds.

#### 3.1.9. Other Compounds

Six other compounds were present in the samples, including 1,3-dioxolane,2,4,5-trimethyl-, 1,3-dioxane, 2-methyl-, naphthalene, 2-methylnaphthalene, 2-phenylthiophene, and 4-acetoxy-3-methoxystyrene. The 4-acetoxy-3-methoxystyrene concentrations were highest in all the vinegar samples with B1, B2, B3, B4, B5, and B6 displaying respective levels of 12,375.86 µg/L, 4469.58 µg/L, 1962.65 µg/L, 1414.54 µg/L, 2734.43 µg/L, and 924.48 µg/L. Furthermore, the naphthalene levels were lowest in all the samples at 1.26 µg/L, 3.48 µg/L, 3.42 µg/L, 0.92 µg/L, 0.79 µg/L, and 27.83 µg/L in B1, B2, B3, B4, B5, and B6, respectively. The concentrations of these compounds did not reach the odor thresholds, contributing little to the aroma of the vinegar.

### 3.2. Aromatic Features of the Chinese Vinegar Samples

The overall aromatic features in the traditional Chinese vinegar samples were assessed according to nine aroma elements, including fruity, floral, herbaceous, nutty, caramel, earthy, chemical, fatty, and roasted. The overall aroma was rated according to the OAVs of each volatile compound that significantly contributed to each aromatic category (OAV above 1). The B1 sample presented sour, green, floral, and sweet scents, while sour, green, fruity, sweet, and roasted aromas were evident in the B2 sample (Table 6). The B3 and B5 samples presented strong sour, green, fruity, and sweet notes, while B4 displayed sour, green, floral, and sweet aromatic notes. The B6 sample presented strong sour, fragrant, green, fruity, and sweet aromas. Besides, the aromatic features of the Chinese vinegar are similar to the other vinegar such as Shanxi aged-vinegar [49], strawberry vinegar [50], and cordyceps vinegar [51].

### 3.3. Aromatic Features of the Chinese Vinegar Samples

The characteristic aromas of different vinegar varieties were analyzed according to the qualitative and quantitative aroma substance results. Figure 1, where 86.9% of the variance is in the first two components, reflects most of the sample information. The results showed that the samples were divided into four groups, with B1 and B4 concentrated in quadrant 2, B2 and B6 concentrated quadrant 3, B5 in quadrant 1, and B3 in quadrant 4.

As shown in Figure 1, acetic acid, caproic acid, butyric acid, and diethyl succinate were located on the positive side of PC1, whereas 3-hydroxy-2-butanone, 4-ethyl-2-methoxyphenol and benzaldehyde were located on the negative side of PC1. Phenethyl alcohol, ethyl benzoate, and 2-methylbutyric acid were located on the positive side of PC2, whereas ethyl acetate, ethyl propionate, and 1-methypyrrole-2-carboxaldehyde were located on the negative side of PC2.

Since the acetic acid, caproic acid, and butyric acid levels in B3 and B5 were high, a positive direction distribution was evident in PC1. Similarly, B4 and B6 displayed distribution in a negative direction in PC1 due to the high 3-hydroxy-2-butanone and benzaldehyde concentrations. Therefore, variation was evident in the characteristic volatile composition of the different vinegar samples.

### 3.4. Analysis of the Overall Consumer Liking

#### 3.4.1. Overall Consumer Liking

This study collected the liking data of 76 qualified consumers. A larger sample size could make the results more accurate and instructive, although our sample size (76) was appropriate for consumer liking, as many studies show. Berna et al. [52], Yanxin et al. [53], and Varela et al. [54] studied tomatoes, Chinese bog bilberry wines, and coffee with a sample size of 54, 93, and 96 respectively. On average, B6 was preferred, receiving a score of 3.60 on a scale of 1 to 7, followed by B2, indicating that the most preferred vinegar was still not liked much (Figure 2). A previous study showed that the satisfaction level of consumers of vinegar products was low at this stage [55]. B1 scored the lowest in liking with a value of 3.18. The liking score range of all the tested products was 0.43, suggesting that respondents provided scores in a relatively limited range. The Kruskal-Wallis test calculated that there was no significant difference between the six products at a significance level of 0.05. This could be attributed to the significant segmentation in the liking results of the respondents, as discussed subsequently. Zamora and Guirao (2004) [56] mentioned that experts gave a more consistent description of attributes than the trained panelists for different wine product brands. 

#### 3.4.2. The Association between the Geographical Location of Consumers for the Past Ten Years and Their Likings

Figure 3 shows a heatmap representing the geographical origins of the respondents and their likings. The results yielded two distinct clusters, one containing B1, B3, and B5, while the other comprising B2, B4, and B6. Respondents from Chongqing, Hunan, and Sichuan generally showed a marked liking for B1, B3, and B5, while participants from Heilongjiang, Henan, Shaanxi, and Inner Mongolia preferred B2, B4, and B6. Respondents from Beijing, Hebei, and Shanxi exhibited lower liking differences regarding the tested products. Moreover, respondents from Hubei, Anhui, and Zhejiang displayed a less positive attitude toward most of the tested samples. Obviously, consumers from different geographical locations had different likings [57].

It should be noted that the respondents were not evenly distributed in this study, denoting an area that could be improved in further research. Here, 48% of the respondents were from Beijing or had lived in Beijing in recent years (Appendix A). This observation may not remain the same when the sample size increases, representing an interesting phenomenon derived from this dataset.

#### 3.4.3. Open Comments from Consumers

The open comments are encompassed in Figure 4a. When describing their liking for the vinegar samples, no significant variation or specific frequency was detected in the verbiage used by the consumers, and included terms, such as “rich vinegar aroma”, “medium vinegar aroma”, “fruity vinegar aroma”, and “sweet aroma.” Consumers also mentioned terms like “too pungent vinegar aroma”, “not rich vinegar aroma”, “alcohol aroma”, and “smelly aroma.” Furthermore, the heatmap showed the differences between the six samples as per the participants.

Regarding liking, B1, B3, and B4 were more often described as presenting a “medium vinegar aroma” than the remaining samples, while more participants ascribed a “rich vinegar aroma” to B2. More consumers described B6 as “having a sweet aroma and fruity vinegar aroma” than the other samples.

Many consumers (32 out of 76) described B5 as exhibiting a “too pungent vinegar aroma”, while significantly fewer participants ascribed this characteristic to B3 and B6, when asked what they disliked about the product. Comments indicated B6 as “having an alcohol aroma” for 15 consumers out of 76, while fewer than five participants ascribe this attribute to the other samples. B3 was attributed a “Chinese medicine aroma” by 15 out 76 consumers, who disliked the characteristics of this sample, representing the highest percentage of the six products.

### 3.5. Consumer Liking Segmentation

#### 3.5.1. Overall Liking by Clusters

The consumers were clustered according to their preference patterns using k-means clustering and PCA, and the results were visualized in a 2-D map (Figure 5). The average likings of the three identified consumer clusters were displayed in the bar plot shown in Figure 6. The clusters contained 30, 18, and 28 participants, respectively. The consumers in cluster 1 generally provided lower liking scores for all the products, with average values below 4. Cluster 1 consumers favored B4 and B6 over the other samples. Cluster 2 consumers preferred B2 and B3, while cluster 3 participants favored the B1, B3, and B6 samples, with liking scores over 4.

#### 3.5.2. Cluster Differences in Demographics, Usage, and Attitude

The three consumer clusters displayed differences in attitude towards purchasing and using vinegar (Figure 7). A higher percentage of cluster 2 consumers considered the product brand essential when purchasing vinegar products than the other two clusters. Cluster 3 consumers often used vinegar as a condiment in daily life when having noodles, while cluster 1 and 2 consumers agreed to a lesser extent in this regard, indicating different vinegar utilization habits. A higher percentage of cluster 1 consumers used rice vinegar than the other clusters, and attached less importance to acidity when purchasing vinegar. A previous study showed that consumers differ in their usage and attitudes towards balsamic vinegar. Italians would pair balsamic vinegar mainly with vegetables, fruits, and cheese, while Koreans would combine balsamic vinegar preferably with bread, vegetables, and meat [58]. In this study, a lot of consumers used vinegar when eating dumplings and noodles.

#### 3.5.3. Open Comments by Clusters

The open comment elicitation rates of each consumer cluster were visualized in a heatmap (Figure 4b). The elicitation rate pattern of the comments showed minimal differences between the clusters, which was validated by the correlation among the groups (results not shown). This consistency suggested that the consumers displayed a limited capability to distinguish and describe the aroma of the vinegar and to express their likes and dislikes.

### 3.6. The Correlation between Consumer Liking and the Volatile Profiles of the Vinegar

PLSR was applied to investigate the correlation between the volatile profiles of the vinegar and the likings of the consumers in each cluster, to reveal the volatile chemicals responsible for favorable aroma scores. The biplot for each cluster is shown in Figure 8. The relative importance of a specific compound was calculated by its percentage of the absolute value of the coefficient in the sum of the absolute value of all coefficients. The top five most important volatiles are listed in Table 7, displaying the different volatile compounds that may drive consumer likings in the clusters. Therefore, 1-methylpyrrole-2-carboxaldehyde, ethyl acetate, acetylfuran, 1H-Pyrrole-2-carbaldehyde, and 2,3,5-trimethylpyrazine played a crucial role in the likings of cluster 1 consumers. Cluster 2 consumers were partial to benzaldehyde, phenylethanal, 3-methyl-1-butanol, 3-hydroxy-2-butanone, and ethyl acetate, while cluster 3 consumers favored products containing (2-methoxy-4-vinyl-phenyl)-acetate, 1,2-propanediol,2-acetate, isobutyl acetate, methylbutyric acid, and isoamyl acetate. Detailed regression coefficients are provided in Appendix A. Jo et al. (2013) [59] reported that the highest score was observed for vinegar with moderate acidity. Cejudo-Bastante et al. (2018) [60] confirmed that for the majority of volatile compounds, higher contents were observed for the submerged culture acetification process, and this was also reflected in the sensory analysis, presenting higher scores for the different descriptors.

## 4. Conclusions

In this study, we used SPME-GC–MS combined with sensory evaluation to examine the association between consumer perception and volatile compounds of six types of Chinese vinegar. The results showed that 52 volatile compounds were detected by GC–MS in six types of Chinese vinegar. High concentrations of ethyl acetate, acetic acid, and phenethyl alcohol were found in all the vinegar samples. Combined with sensory evaluation, it was found that some specific volatile compounds affected consumer liking for Chinese vinegar significantly. The potato vinegar (B6) was preferred; more consumers described B6 as “having a sweet aroma and fruity vinegar aroma” than the other samples, and many consumers described the Heng-shun Jinyou balsamic vinegar (B5) as exhibiting a “too pungent vinegar aroma” when asked what they disliked about the product. For PLSR, the most important volatile compounds in the three clusters that drove consumer liking confirmed the importance of 1-methylpyrrole-2-carboxaldehyde, ethyl acetate, acetylfuran, 1H-Pyrrole-2-carbaldehyde, 2,3,5-trimethylpyrazine, benzaldehyde, phenylethanal, 3-methyl-1-butanol, 3-hydroxy-2-butanone, (2-methoxy-4-vinyl-phenyl)-acetate, 1,2-propanediol,2-acetate, isobutyl acetate, methylbutyric acid, and isoamyl acetate in Chinese vinegar. Manufacturers should pay attention to the changes in these 14 compounds and the content of the end product in the production process, and at the same time accumulate data about the correlation between compounds and consumer liking. They should subsequently change the production process using the data to improve the quality and consumer preference of vinegar.

## Figures and Tables

**Figure 1 foods-11-02224-f001:**
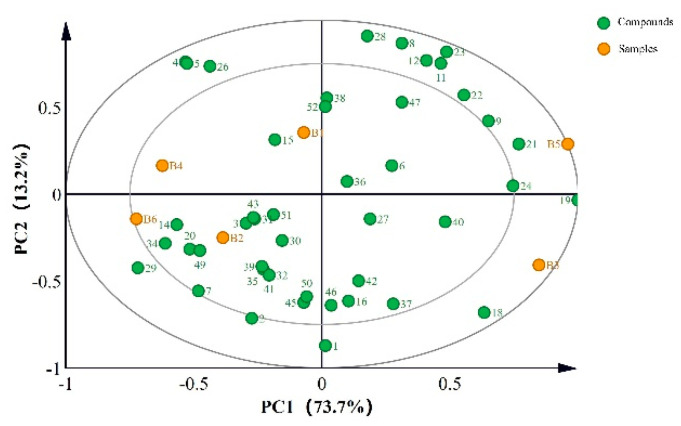
Principal component analysis model of the key aroma compounds in different vinegar samples. The sample codes correspond with Table 1. The number of compounds corresponds with Table 2.

**Figure 2 foods-11-02224-f002:**
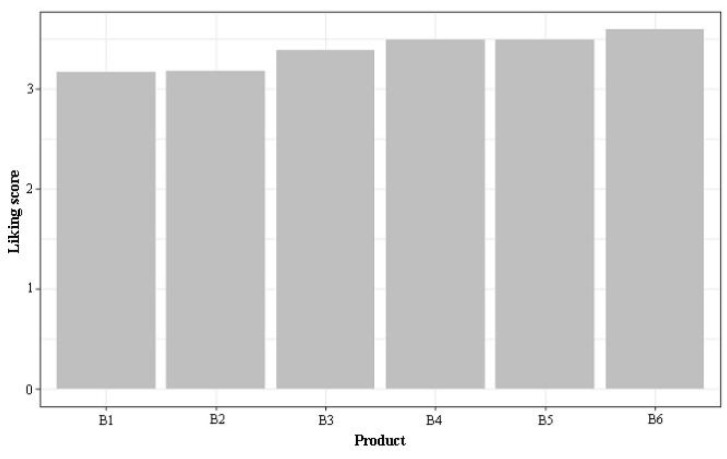
Average liking scores of 76 qualified consumers for six kinds of vinegar. The vinegar code corresponds with Table 1.

**Figure 3 foods-11-02224-f003:**
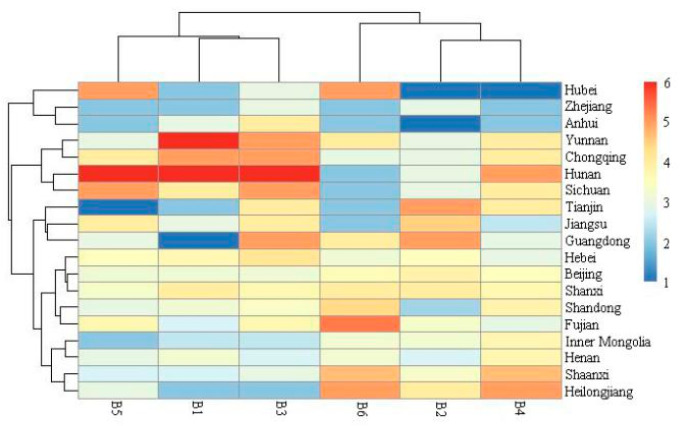
The heatmap of respondents’ liking scores for six kinds of vinegar in different regions. The code of vinegar corresponds with Table 1. These regions are explained in the Appendix A.

**Figure 4 foods-11-02224-f004:**
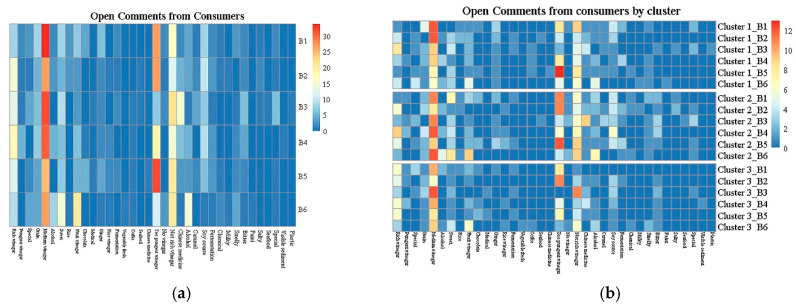
The heatmaps of open comments from all consumers (**a**) and each cluster (**b**). The code of vinegar corresponds with Table 1. The left 18 columns were the terms that consumers liked, and the right 18 columns were the terms that consumers disliked.

**Figure 5 foods-11-02224-f005:**
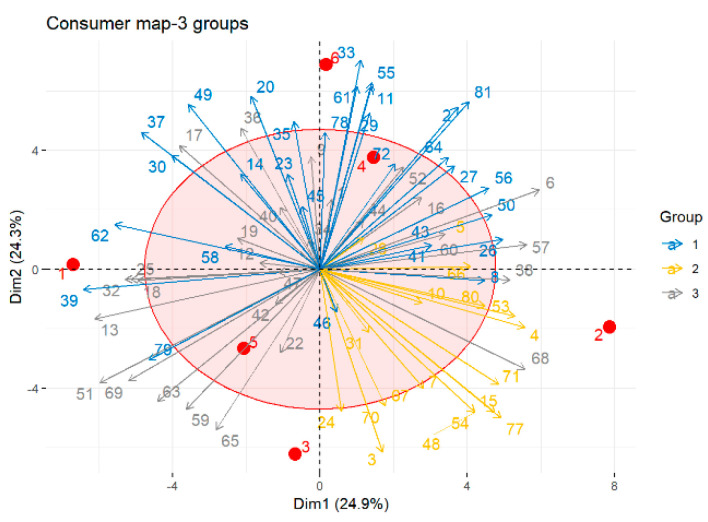
The 2-D map of the consumer liking pattern according to their preference patterns using k-means clustering and PCA.

**Figure 6 foods-11-02224-f006:**
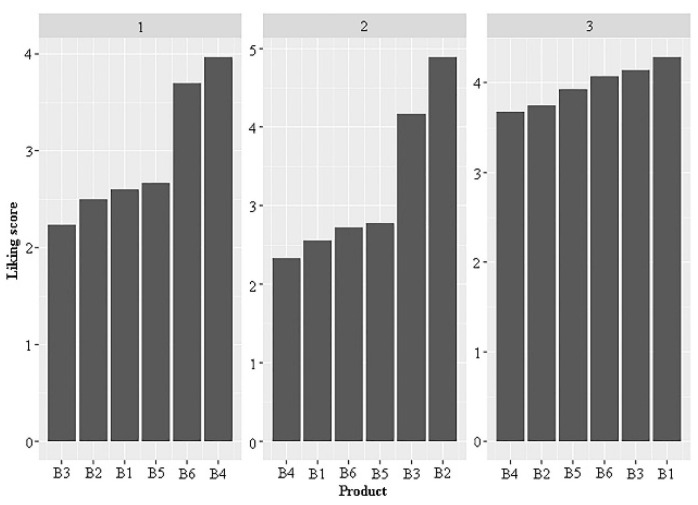
The bar plot of average liking scores by three clusters of consumers for products (n = 30 for cluster 1, n = 18 for cluster 2 and n = 28 for cluster 3). The vinegar code corresponds with Table 1.

**Figure 7 foods-11-02224-f007:**
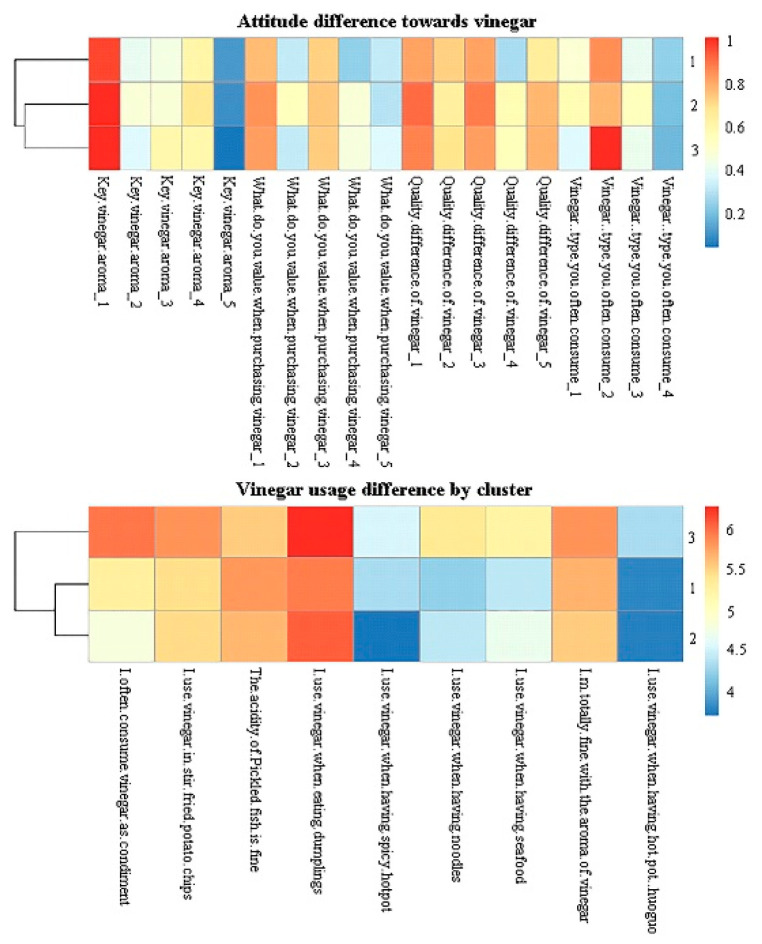
The heatmap of three clusters of consumers reflecting attitudes towards purchasing and using vinegar (n = 30 for cluster 1, n = 18 for cluster 2 and n = 28 for cluster 3).

**Figure 8 foods-11-02224-f008:**
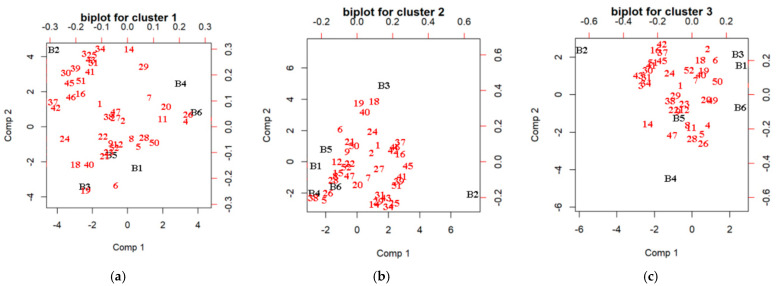
The biplot of each cluster (**a**–**c**) by partial least squares regression analysis (n = 30 for cluster 1, n = 18 for cluster 2 and n = 28 for cluster 3). The vinegar code corresponds with Table 1. The number of compounds corresponds with Table 2.

**Table 1 foods-11-02224-t001:** Detailed information on six kinds of Chinese vinegar.

Code	B1	B2	B3	B4	B5	B6
**Product name**	Ten-year aged Qian-he cellar vinegar	Ning-hua-mansion old vinegar	East-lake health vinegar	Qian-he glutinous rice vinegar	Heng-shun Jinyou balsamic vinegar	Potato vinegar
**Net content** **(mL)**	500	500	500	500	360	1750
**Ingredients**	Water, glutinous rice, wheat, sorghum, corn, buckwheat, edible salt, sugar	Drinking water, sorghum, bran, rice husk, daqu (barley, peas), edible salt, spices	Water, sorghum, barley, peas, honey, dates, peanuts, licorice, hawthorn, sugar	Water, glutinous rice, rice, wheat bran, edible salt, sugar	Water, glutinous rice, wheat bran, edible salt, sugar	Drinking water, potato, edible salt, food additive (sodium benzoate)
**Product standard No**	GB/18187SSF	GB/18187SSF	GB19777SSF	GB/T18187SSF	GB/T18187SSF	GB/T18187SSF
**Goods origin**	Meishan, Sichuan Province	Taiyuan, Shanxi Province	Taiyuan, Shanxi Province	Meishan, Sichuan Province	Zhenjiang, Jiangsu Province	Ulanqab, Inner Mongolia
**Product type**	Mature vinegar	Mature vinegar	Health vinegar	Aromatic vinegar	Aromatic vinegar	Potato vinegar

**Table 2 foods-11-02224-t002:** Relevant information of gas chromatography–mass spectrometry (GC–MS) analysis on vinegar compounds.

No.	Compound	RI	Method of Identification	Class	Quantitative Ion	Aroma Description	Odor Threshold (μg/L)
1	ethyl acetate	878	MS, RI	Ester	43	sweet, etheric, fruity, grape, rum	5000
2	ethyl propionate	958	MS, RI	Ester	75	sweet, fruity, grape, ether, rum, pineapple	10
3	n-propyl acetate	974	MS, RI	Ester	43	pleasant, solvent, sweet fruit	150,000
4	isobutyl acetate	1012	MS, RI	Ester	43	sweet, apple, banana, fruity	66
5	isoamyl acetate	1110	MS, RI	Ester	43	sweet, pear, banana, fruity	—
6	1,2-propanediol,2-acetate	1621	MS, RI	Ester	43	—	—
7	trimethylene acetate	1665	MS, RI	Ester	43	—	11
8	ethyl benzoate	1681	MS, RI	Ester	105	sweet, fruity, fragrant	60
9	diethyl succinate	1680	MS, RI	Ester	101	weak pleasing aroma	2000
10	ethyl phenylacetate	1770	MS, RI	Ester	91	sweet, fruity, cocoa, floral scent, honey aroma	650
11	β-phenethyl acetate	1788	MS, RI	Ester	104	sweet, green, floral, fruity, citrus, honey	3900
12	3-methylbutyraldehyde	924	MS, RI	Aldehyde	44	apple, chocolate, cocoa	—
13	nonanal	1386	MS, RI	Aldehyde	57	fat, floral, waxy, citrus	1
14	benzaldehyde	1520	MS, RI	Aldehyde	77	almonds, cherries, nuts, woody	3500
15	phenylethanal	1638	MS, RI	Aldehyde	91	green, earthy, chocolate	4
16	1H-pyrrole-2-carbaldehyde	2009	MS, RI	Aldehyde	95	—	—
17	5-methyl-2-phenyl-2-hexenal	2060	MS, RI	Aldehyde	117	bitter cocoa, nuts, honey, baking and grassy notes	—
18	1-methylpyrrole-2-carboxaldehyde	2009	MS, RI	Aldehyde	95	—	—
19	acetic acid	1449	MS, RI	Acid	60	strong sour taste	2200
20	propionic acid	1522	MS, RI	Acid	74	spicy and sour	20,000
21	butyric acid	1620	MS, RI	Acid	60	cheese, milk, cream, fruity	240
22	isovaleric acid	1680	MS, RI	Acid	60	cheese, products, fruity	700
23	2-methylbutyric acid	1685	MS, RI	Acid	74	pungent and spicy Roquefort	20
24	caproic acid	1880	MS, RI	Acid	60	green, woody, grassy, vegetable, meaty, fruity	3000
25	octanoic acid	2100	MS, RI	Acid	60	sweet	3000
26	3-methyl-1-butanol	1185	MS, RI	Alcohol	55	apple, banana, whiskey	30,000
27	2,3-butanediol	1584	MS, RI	Alcohol	45	sweet, butter, butter	100,000
28	phenethyl alcohol	1890	MS, RI	Alcohol	91	sweet, green, floral, fresh bread aroma	750
29	3-hydroxy-2-butanone	1270	MS, RI	Ketone	45	—	140
30	acetophenone	1656	MS, RI	Ketone	105	cream, fat	65
31	2-pyrrolidinone	2037	MS, RI	Ketone	42	strong medicinal, almond	—
32	guaiacol	1862	MS, RI	Phenol	109	smoked, spicy, fragrant, meaty, woody	21
33	2-ethyl-3-hydroxy-4H-pyran-4-one	2052	MS, RI	Phenol	140	fruity, caramel	—
34	4-ethyl-2-methoxyphenol	2031	MS, RI	Phenol	137	sweet, spicy, herbal	—
35	4-ethylphenol	2199	MS, RI	Phenol	107	strong phenolic smell, slightly sweet aroma	—
36	furfural	1462	MS, RI	Furan	96	strong phenolic smell, slightly sweet aroma	3000
37	acetylfuran	1508	MS, RI	Furan	95	baked incense, smoky	10,000
38	furfuryl acetate	1525	MS, RI	Furan	81	ester, floral	—
39	1-pentanone, 1-(2-furanyl)-	1563	MS, RI	Furan	95	—	6
40	3-furanmethanol	1679	MS, RI	Furan	98	caramel	—
41	1-(5-methyl-2-furyl)ethan-1-one	1606	MS, RI	Furan	109	biscuits, roasted almonds	—
42	4-(2-furyl)-3-buten-2-one	1879	MS, RI	Furan	121	sweet, powdery, nutty, creamy, woody cinnamon	—
43	5-acetyldihydrofuran-2(3H)-one	2160	MS, RI	Furan	85	sweet, lemon green	65
44	2-methylpyrazine	1266	MS, RI	Pyrazine	94	nuts, peanuts, roasted incense, soily, mildew	60
45	2,3-dimethyl pyrazine	1356	MS, RI	Pyrazine	67	mildew, roasted, creamy, nuts, cocoa, coffee	2500
46	2,3,5-trimethylpyrazine	1415	MS, RI	Pyrazine	42	baked potatoes, fried peanuts, nuts, earthy notes, fermented	400
47	1,3-dioxolane,2,4,5-trimethyl-	967	MS, RI	Others	44	—	—
48	1,3-dioxane, 2-methyl-	1044	MS, RI	Others	87	—	—
49	naphthalene	1744	MS, RI	Others	128	aromatic odor, coal tar smell	1500
50	2-methylnaphthalene	1839	MS, RI	Others	142	aromatic odor, coal tar, camphor, chemicals	—
51	2-phenylthiophene	2124	MS, RI	Others	160	—	—
52	4-acetoxy-3-methoxystyrene	2235	MS, RI	Others	150	—	—

MS: mass spectra; RI: retention indice; Both agreed with database of NIST11; Odor thresholds were from the literature [32].

**Table 3 foods-11-02224-t003:** Relevant information of the standard curve on vinegar compounds.

No.	Compound	Class	Purity	Supplier	Linear Range(μg/L) LOQ/LOD	Formula	R^2^
1	ethyl acetate	Ester	0.998	Sigma-Aldrich	121,137/40,379	y = 30.114x − 191,619	R^2^ = 0.9811
5	isoamyl acetate	Ester	0.95	Sigma-Aldrich	433.125/144.375	y = 0.0102x − 31.932	R^2^ = 0.918
9	diethyl succinate	Ester	0.99	Sigma-Aldrich	99.9375/33.3125	y = 0.0081x − 4.7535	R^2^ = 0.9829
14	benzaldehyde	Aldehyde	0.99	Sigma-Aldrich	300/100	y = 0.0068x − 188.12	R^2^ = 0.9911
20	propionic acid	Acid	0.995	Sigma-Aldrich	6135/2045	y = 0.5732x + 1246.3	R^2^ = 0.9697
22	isovaleric acid	Acid	0.99	Sigma-Aldrich	116.25/38.75	y = 0.0012x + 13.158	R^2^ = 0.9939
24	caproic acid	Acid	0.995	Sigma-Aldrich	2437.5/812.5	y = 0.0911x + 507.9	R^2^ = 0.9661
25	octanoic acid	Acid	0.99	Sigma-Aldrich	6761.25/2253.75	y = 0.0438x + 902	R^2^ = 0.9837
28	phenethyl alcohol	Alcohol	0.99	Sigma-Aldrich	30,825/10,275	y = 0.1457x − 18,131	R^2^ = 0.9953

**Table 4 foods-11-02224-t004:** Content information of vinegar compounds (μg/L).

No.	B1 ^a^	B2	B3	B4	B5	B6
1	130,650.99 ± 6636.18 ^a^	625,514.35 ± 1291.87 ^e^	783,331.79 ± 6806.85 ^f^	571,951.58 ± 3619.70 ^d^	304,167.86 ± 5632.55 ^b^	467,917.75 ± 8154.52 ^c^
2	1181.77 ± 4.53 ^b^	1704.31 ± 14.55 ^c^	1734.33 ± 10.94 ^c^	221.56 ± 8.28 ^a^	223.86 ± 4.31 ^a^	2271.78 ± 47.05 ^d^
3	751.54 ± 38.69 ^a^	5244.03 ± 76.36 ^c^	399.66 ± 23.07 ^a^	2123.75 ± 34.38 ^b^	1792.44 ± 61.09 ^b^	812.39 ± 536.96 ^a^
4	740.49 ± 7.53 ^f^	126.34 ± 1.17 ^b^	16.24 ± 0.18 ^a^	637.52 ± 6.18 ^e^	278.41 ± 2.77 ^c^	472.90 ± 30.91 ^d^
5	21,239.27 ± 78.78 ^d^	3100.82 ± 47.47 ^a^	tr	22,149.40 ± 183.22 ^e^	7331.81 ± 70.42 ^b^	11,597.80 ± 2340.62 ^c^
6	344,314.42 ± 3037.80 ^d^	8624.00 ± 193.86 ^a b^	208,404.31 ± 1309.92 ^c^	6464.71 ± 627.42 ^a^	8724.21 ± 206.27 ^a b^	11,842.14 ± 825.47 ^b^
7	159.27 ± 6.92 ^a^	1357.87 ± 113.19 ^c^	761.25 ± 39.64 ^b^	138.21 ± 18.20 ^a^	152.63 ± 2.70 ^a^	2898.76 ± 2.70 ^d^
8	2496.29 ± 5.54 ^d^	768.78 ± 3.38 ^c^	57.64 ± 15.06 ^a^	2766.54 ± 4.32 ^e^	4425.84 ± 3.75 ^f^	134.63 ± 36.13 ^b^
9	607.73 ± 1.70 ^a^	1786.03 ± 57.56 ^c^	1354.23 ± 92.26 ^b^	1881.49 ± 200.04 ^d^	11,395.23 ± 347.24 ^e^	tr
10	tr	tr	tr	tr	tr	tr
11	0.45 ± 0.19 ^b^	tr	tr	0.62 ± 0.16 ^b^	1.47 ± 0.30 ^c^	tr
12	39,262.87 ± 58.38 ^e^	22,572.34 ± 19.84 ^c^	19,246.37 ± 27.46 ^b^	35,396.43 ± 20.06 ^d^	44,409.15 ± 95.67 ^f^	3022.45 ± 337.71 ^a^
13	tr	tr	tr	tr	tr	tr
14	1150.98 ± 2.29 ^a^	6667.33 ± 10.79 ^e^	2167.45 ± 12.54 ^b^	7281.14 ± 0.96 ^f^	2704.11 ± 11.99 ^c^	3383.24 ± 156.20 ^d^
15	16,618.10 ± 34.91 ^c^	13,683.19 ± 2.56 ^b^	16,136.22 ± 62.54 ^c^	23,161.05 ± 76.12 ^d^	14,106.61 ± 38.35 ^b^	12,303.69 ± 556.86 ^a^
16	1982.62 ± 135.83 ^b^	14,611.62 ± 877.44 ^d^	7735.74 ± 1024.88 ^c^	1439.58 ± 89.46 ^a b^	2466.34 ± 16.66 ^b^	223.47 ± 7.66 ^a^
17	tr	tr	tr	tr	tr	tr
18	36.40 ± 2.62 ^a^	135.69 ± 7.68 ^a^	489.55 ± 108.62 ^b^	27.11 ± 1.55 ^a^	92.43 ± 2.28 ^a^	14.36 ± 2.46 ^a^
19	8,020,749.17 ± 2,250,142.09 ^a^	7,057,979.52 ± 1,329,019.25 ^a^	11,429,688.79 ± 3,499,562.07 ^a^	6,119,026.91 ± 530,020.51 ^a^	11,729,001.22 ± 5,750,084.35 ^a^	6,077,922.73 ± 124,119.42 ^a^
20	13,954.14 ± 125.32 ^a^	83,294.53 ± 427.16 ^c^	24,429.37 ± 153.02 ^b^	tr	tr	503,640.81 ± 153,772.98 ^d^
21	73.83 ± 0.16 ^a b^	155.64 ± 0.25 ^b c^	230.41 ± 35.27 ^c^	46.56 ± 0.31 ^a^	882.92 ± 73.08 ^d^	40.22 ± 0.13 ^a^
22	363.08 ± 0.08 ^c^	397.24 ± 17.93 ^c^	246.81 ± 0.38 ^b^	379.70 ± 23.62 ^c^	793.42 ± 0.05 ^d^	137.08 ± 23.54 ^a^
23	13,432.09 ± 390.75 ^d^	8349.21 ± 731.17 ^b c^	6811.15 ± 137.18 ^a b^	9544.95 ± 925.82 ^c^	16,317.27 ± 1529.55 ^e^	5065.04 ± 633.79 ^a^
24	11,979.79 ± 5.92 ^c^	21,522.27 ± 328.98 ^e^	17,502.03 ± 33.56 ^d^	8781.48 ± 980.37 ^b^	28,541.34 ± 986.88 ^f^	4208.75 ± 13.15 ^a^
25	tr	tr	tr	tr	tr	tr
26	8637.39 ± 32.20 ^c^	440.01 ± 39.83 ^a^	tr	13,010.93 ± 167.42 ^d^	4574.55 ± 42.91 ^b^	4819.93 ± 1259.78 ^b^
27	44,617.93 ± 137.00 ^a^	95,440.55 ± 28,644.95 ^b c^	49,629.19 ± 601.14 ^a b^	20,513.18 ± 528.50 ^a^	106,686.91 ± 181.98 ^c^	92,842.65 ± 31,051.35 ^b c^
28	460,480.92 ± 5729.13 ^c^	145,519.63 ± 11,269.22 ^a b^	77,189.78 ± 7544.10 ^a^	565,184.73 ± 42,979.58 ^d^	695,627.22 ± 40,986.21 ^e^	173,905.87 ± 24,280.65 ^b^
29	tr	6910.25 ± 1724.70 ^b c^	136.94 ± 133.61 ^a b^	2468.94 ± 521.10 ^a b c^	369.80 ± 283.54 ^a b^	9315.87 ± 5955.46 ^c^
30	1412.94 ± 5.75 ^b^	5653.94 ± 331.67 ^c^	1057.34 ± 9.49 ^b^	1257.21 ± 9.80 ^b^	1220.89 ± 5.24 ^b^	129.10 ± 1.78 ^a^
31	285.25 ± 29.00 ^a b^	769.83 ± 101.22 ^c^	185.04 ± 24.28 ^a^	328.93 ± 13.26 ^b^	362.42 ± 23.59 ^b^	250.98 ± 57.63 ^a b^
32	1385.18 ± 5.38 ^a^	42,367.62 ± 1490.59 ^c^	5063.67 ± 6.19 ^b^	593.83 ± 117.46 ^a^	809.65 ± 31.79 ^a^	1260.93 ± 244.56 ^a^
33	tr	tr	tr	tr	tr	tr
34	1836.67 ± 125.52 ^b^	10,330.78 ± 422.46 ^e^	413.50 ± 63.52 ^a^	4746.59 ± 87.82 ^d^	199.59 ± 15.38 ^a^	2794.37 ± 656.28 ^c^
35	23.19 ± 1.27 ^a^	2829.93 ± 226.93 ^b^	180.21 ± 15.24 ^a^	14.55 ± 0.59 ^a^	18.91 ± 0.36 ^a^	30.89 ± 6.62 ^a^
36	334.34 ± 3.85 ^c^	1502.79 ± 54.64 ^f^	142.65 ± 11.07 ^b^	514.31 ± 4.86 ^d^	1057.21 ± 8.29 ^e^	tr
37	4810.83 ± 49.30 ^b^	24,330.76 ± 282.79 ^e^	17,231.94 ± 407.93 ^d^	5400.72 ± 58.23 ^b^	9834.93 ± 152.36 ^c^	3161.14 ± 549.63 ^a^
38	19,300.71 ± 1153.50 ^e^	15,329.97 ± 1124.08 ^d^	8932.46 ± 178.69 ^b^	18,218.52 ± 51.99 ^e^	12,829.97 ± 133.62 ^c^	965.43 ± 193.69 ^a^
39	93.85 ± 0.52 ^b^	1474.57 ± 75.16 ^c^	122.23 ± 11.09 ^b^	46.35 ± 0.08 ^a b^	2.39 ± 0.45 ^a^	tr
40	1274.10 ± 0.29 ^c^	787.66 ± 50.90 ^b^	2064.35 ± 128.89 ^d^	1293.57 ± 93.83 ^c^	711.91 ± 0.28 ^b^	tr
41	135.85 ± 0.72 ^a^	4078.41 ± 326.13 ^c^	502.77 ± 0.71 ^b^	64.87 ± 5.68 ^a^	84.82 ± 9.87 ^a^	124.02 ± 24.02 ^a^
42	785.24 ± 9.04 ^d^	1809.90 ± 16.33 ^f^	1156.85 ± 3.01 ^e^	323.94 ± 8.20 ^b^	393.02 ± 26.61 ^c^	93.58 ± 38.26 ^a^
43	479.21 ± 43.10 ^a b^	1265.54 ± 185.14 ^c^	303.70 ± 35.28 ^a^	544.40 ± 19.80 ^b^	592.24 ± 33.21 ^b^	413.46 ± 82.58 ^a b^
44	tr	tr	tr	tr	tr	tr
45	297.71 ± 2.71 ^b^	3190.07 ± 127.80 ^e^	1498.79 ± 10.60 ^d^	627.20 ± 83.01 ^c^	91.46 ± 13.13 ^a^	45.30 ± 4.69 ^a^
46	29,125.40 ± 294.08 ^b^	114,532.53 ± 1137.68 ^e^	94,445.64 ± 2367.40 ^d^	53,390.76 ± 5265.10 ^c^	6870.70 ± 140.14 ^a^	911.37 ± 106.41 ^a^
47	1780.01 ± 3.98 ^c^	3373.76 ± 15.08 ^d^	1324.12 ± 9.50 ^b^	5712.76 ± 55.65 ^e^	8207.16 ± 33.79 ^f^	1020.24 ± 43.86 ^a^
48	tr	tr	tr	tr	tr	tr
49	1.26 ± 0.01 ^a^	3.48 ± 0.01 ^a^	3.42 ± 1.87 ^a^	0.92 ± 0.21 ^a^	0.79 ± 0.12 ^a^	27.83 ± 10.31 ^b^
50	70.87 ± 0.62 ^a b^	120.71 ± 1.83 ^a b^	612.06 ± 342.68 ^b c^	51.49 ± 3.51 ^a^	161.21 ± 37.54 ^a b^	924.48 ± 344.18 ^c^
51	1172.28 ± 1876.58 ^a^	2650.75 ± 1886.25 ^a^	323.81 ± 252.06 ^a^	324.17 ± 494.20 ^a^	829.70 ± 296.07 ^a^	512.85 ± 12.92 ^a^
52	12,375.86 ± 1661.42 ^c^	4469.58 ± 1100.56 ^b^	1962.65 ± 378.17 ^a^	1414.54 ± 16.80 ^a^	2734.43 ± 312.57 ^a b^	756.57 ± 157.68 ^a^

“tr” represents “Trace Amount”. The data are mean ± standard deviation of one-way ANOVA and Duncan’s range test. The different letters in each row indicate significant differences at a significant level of 0.05. The code of vinegar corresponds with Table 1. The number of compounds corresponds with Table 2.

**Table 5 foods-11-02224-t005:** Odor activity values (OAVs) of vinegar compounds.

No.	B1	B2	B3	B4	B5	B6
1	26.130	125.103	156.666	114.390	60.834	93.584
2	118.177	170.431	173.433	22.156	22.386	227.178
4	11.220	1.914	—	9.659	4.218	7.165
7	14.479	123.443	69.205	12.564	13.876	263.524
8	41.605	12.813	—	46.109	73.764	2.244
9	—	—	—	—	5.698	—
14	—	1.905	—	2.080	—	—
15	4154.525	3420.799	4034.055	5790.261	3526.653	3075.922
19	3645.795	3208.173	5195.313	2781.376	5331.364	2762.692
20	—	4.165	1.221	—	—	25.182
21	—	—	—	—	3.679	—
22	—	—	—	—	1.133	—
23	191.887	119.274	97.302	136.356	233.104	72.358
24	3.993	7.174	5.834	2.927	9.514	1.403
27	—	—	—	—	1.067	—
28	613.975	194.026	102.920	753.580	927.503	231.874
29	—	49.359	—	17.635	2.641	66.542
30	21.738	86.984	16.267	19.342	18.783	1.986
32	65.961	2017.506	241.127	28.278	38.555	60.044
37	—	2.433	1.723	—	—	—
39	15.642	245.762	20.371	7.726	—	—
43	7.372	19.470	4.672	8.375	9.111	6.361
46	16.181	63.629	52.470	29.662	3.817	—

The number of compounds corresponds with Table 2. The code of vinegar corresponds with Table 1.

**Table 6 foods-11-02224-t006:** Aromatic features of six kinds of vinegar.

Produce Name	Aroma Description
Ten-year aged Qian-he cellar vinegar	Sour, green, floral, and sweet scents
Ning-hua-mansion old vinegar	Sour, green, fruity, sweet, and roasted aromas
East-lake health vinegar	Sour, green, fruity, and sweet notes
Qian-he glutinous rice vinegar	Sour, green, floral, and sweet aromatic notes
Heng-shun Jinyou balsamic vinegar	Sour, green, fruity, and sweet notes
Potato vinegar	Sour, fragrant, green, fruity, and sweet aromas

**Table 7 foods-11-02224-t007:** The relative importance of the top five most important volatiles in three clusters.

Cluster 1	Relative Importance	Cluster 2	Relative Importance	Cluster 3	Relative Importance
1-methylpyrrole-2-carboxaldehyde	4.342935	benzaldehyde	7.217879	4-acetoxy-3-methoxystyrene	8.302227
ethyl acetate	4.319396	phenylethanal	4.1944	1,2-propanediol,2-acetate	8.09123
acetylfuran	4.269728	3-methyl-1-butanol	4.082279	isobutyl acetate	3.474666
1H-pyrrole-2-carbaldehyde	3.784573	3-hydroxy-2-butanone	3.887411	2-methylbutyric acid	3.007397
2,3,5-trimethylpyrazine	3.509161	ethyl acetate	3.391669	isoamyl acetate	2.51753

## Data Availability

Data is contained within the article or Appendix A.

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
