# Peer review of "Study of Consumer Liking of Six Chinese Vinegar Products and the Correlation between These Likings and the Volatile Profile"

_foods, 2022, doi:10.3390/foods11152224_

Round 1
Reviewer 1 Report
Title
The title suggests that a correlation was done between consumer preferences and volatile profile. However, consumer liking is what was determined and then correlated with volatile compounds. The title should be revised to reflect what was done.
Abstract
The Authors have indicated that in this study, a combination of sensory data and instrumental measurements were performed to understand the flavor differences of six types of Chinese vinegar, and then guide manufacturers to improve the product quality and the satisfaction of the customer.
What guidance has been given to manufacturers to improve the product quality and the satisfaction of the consumer? At least such guidance should be highlighted in the abstract since it is one of the main goals or significance of this study.
The use of preference and flavor in the abstract, and in subsequent sections, in general, is misleading. In sensory science, preference and liking are different. One may prefer a product over the other, but it does not imply that the preferred product is liked. Likewise, flavor and aroma are different. Consumers in this study evaluated the aroma of the samples not flavor. Flavor is a combination of taste, aroma, and chemical feelings.
Introduction
Well written except for the paragraph starting from lines 67 to 76. There is no clear connection between the cited literature (Lines 69 – 76) to this study.
Methods
The chemical analysis method is well described, and the use of external and internal standards was good. Likewise, the inclusion of alkanes to calculate retention indices.
Line 111, replace ‘was listed …’ with ‘is listed …’
Table 1, under the code column, ‘produce name’ should be ‘product name’
Line 116, should read, …. sample (5 mL) was mixed with 1 µL of 0.816 µg/µL 2-methyl-3-heptanone ….
In section 2.4, temperatures should be written properly as you have done in section 2.3
In section 2.5, the authors did not describe in detail the methodology for determining OAV. The authors should describe how the perception threshold was determined and how the comparison with concentrations of volatile compounds was done.
Line 180, what was the actual temperature of the samples? How was the temperature kept constant throughout the test and for all the consumers?
A Likert scale has a neutral point (Neither Like nor Dislike). In this case, is the ‘Just so so’ the neutral point?
Line 157 shows that 86 consumers participated while in line 428, it seems 76 consumers participated. Which is the correct number of participants?
Line 198, ANOVA was used to analyze consumer liking data as well? Did the data meet all the needed assumptions e.g., normal distribution, equal variance etc?
Line 204, why was k-means clustering used and not hierarchical clustering? You knew the number of clusters you wanted already? Was the sample size (76) appropriate for cluster analysis?
Results
In sections 3.4 and 3.5, the use of both preference and liking is confusing. The scale used was measuring overall liking. Be consistent.
In figure 2, the label for Y-axis should be liking score. Do Likewise for Figure 6.
Figures or Table captions should be complete (stand-alone) without referring readers to other Tables for interpretation of codes. For instance, in Figure 2, meanings for BI, B2, B3, B4, B5, B6 should be given without referring the reader to Table 1. Do this for all Figures and Tables where applicable.
Clearly missing is a proper and adequate discussion of the findings. Section 3, as rightly indicated is just for results.
Indicate limitations of the study, especially on sample size used for consumer liking.
Conclusions
Give clear guidance to manufacturers to improve the quality and consumer preference of vinegar since this was one of the goals of this study. As it is, it is not obvious.

Reviewer 2 Report
The experimental workflow is well described as well as the statistical evaluation of the results.
I have only a question and one annotation:
How did you create the calibration curves? By GC-MS or GC-FID?
You must add at least one GC-MS chromatogram with peak identification according to the numbers inserted in table 1.
LOQ (the lowest point of the linear range) and LOD must be calculated and added to the method validation table, and at least reported in the main manuscript.
